# Total Culturable Microbial Diversity of Food Contact Surfaces in Poultry and Fish Processing Industries After the Pre-Operational Cleaning Process

**DOI:** 10.3390/foods14132387

**Published:** 2025-07-06

**Authors:** Luiz Gustavo Bach, Gabriela Zarpelon Anhalt Braga, Márcia Cristina Bedutti, Layza Mylena Pardinho Dias, Emanoelli Aparecida Rodrigues dos Santos, Leonardo Ereno Tadielo, Evelyn Cristine da Silva, Jhennifer Arruda Schmiedt, Virgínia Farias Alves, Elaine Cristina Pereira De Martinis, Fábio Sossai Possebon, Vinicius Cunha Barcellos, Luciano dos Santos Bersot

**Affiliations:** 1Department of Veterinary Sciences, Palotina Campus, Federal University of Paraná (UFPR), Rua Pioneiro, 2153, Jardim Dallas, Palotina 85950-000, PR, Brazil; lugubach@hotmail.com (L.G.B.); gabriela_anhalt@hotmail.com (G.Z.A.B.); mcbedutti@gmail.com (M.C.B.); layzamylena85@gmail.com (L.M.P.D.); jhenni_as@hotmail.com (J.A.S.); lucianobersot@ufpr.br (L.d.S.B.); 2School of Veterinary Medicine and Animal Science, Botucatu Campus, São Paulo State University (UNESP), Distrito de Rubião Jr, SN, Botucatu 18618-681, SP, Brazil; emanoelli.santos@unesp.br (E.A.R.d.S.); evelyn.cristine@unesp.br (E.C.d.S.); fabio.possebon@unesp.br (F.S.P.); 3Department of Animal Production and Food, State University of Santa Catarina, Lages 88040-900, SC, Brazil; leonardoerenotadielo@gmail.com; 4Faculdade de Farmácia, Universidade Federal de Goiás, Goiânia 74605-170, GO, Brazil; virginia_alves@ufg.br; 5Ribeirão Preto School of Pharmaceutical Sciences, University of São Paulo, Av. Prof. Dr. Zeferino Vaz, Vila Monte Alegre, Ribeirão Preto 14040-900, SP, Brazil

**Keywords:** biofilms, environmental monitoring, fish slaughtering, poultry processing microbial ecology, high-throughput DNA sequencing

## Abstract

This study assessed the viable and culturable microbial diversity that remained on equipment surfaces after hygiene procedures in Brazilian poultry and fish slaughterhouses. Food-contact surface samples were collected using sterile swabs in poultry (*n* = 50) and fish (*Oreochromis niloticus*, *n* = 50) slaughterhouses. The swab samples were used to prepare culture plates to recover viable and culturable cells. The grown plates were washed, and the total DNA of the cell suspension was extracted with a commercial kit. Sequencing of the total DNA extracted from cultures was targeted at the V3 and V4 regions of the *16S rRNA*. DNA reads were analyzed by QIIME2 software, with results expressed in relative frequency (%RF). Alpha and beta diversity indexes were analyzed considering the spots of sample collection, type of industry, surfaces (smooth or modular), and materials (polypropylene, stainless steel, or polyurethane). The results showed that in the poultry slaughterhouse, the most abundant genera were *Acinetobacter* (27.4%), *Staphylococcus* (7.7%), and *Pseudomonas* (5.3%), while for the fish slaughterhouse, there was a higher abundance of *Staphylococcus* (27.7%), *Acinetobacter* (17.2%), and *Bacillus* (12.5%). Surface characteristics influenced the microbial diversity, with *Acinetobacter* spp. dominating modular surfaces and *Staphylococcus* spp. prevailing on smooth surfaces. The results obtained indicate there is an important resident microbiota that persists even after hygiene processes, and surface-specific cleaning strategies should be developed.

## 1. Introduction

Slaughterhouses are environments that need to be carefully monitored due to potential contamination by pathogenic and/or spoilage microorganisms, which represent a threat to public health and product quality [1,2,3].

To this end, Food Safety Management Systems (FSMSs) are applied, which are a set of processes, procedures, and tools, based on standards and regulations, developed to control hazards and ensure compliance with food safety requirements. These systems cover, among other things, prerequisite programs such as Good Hygiene Practices (GHPs) and Sanitation Standard Operating Procedures (SSOPs) [4,5].

In this scenario, the implementation of SSOPs is key to preventing foodborne diseases since it recommends routine pre-operational cleaning and sanitation [6,7]. Although these initial steps can significantly reduce microbial loads to acceptable levels, the total eradication of contaminants from processing environments is challenging due to several factors: (i) complex equipment design; (ii) different materials of food-contact and non-food-contact surfaces; (iii) extrinsic conditions, such as temperature and humidity; (iv) choice of sanitizers; (v) composition of food matrices or debris; and (vi) microbial diversity [8,9].

Industries usually assess the effectiveness of SSOPs by monitoring indicator microorganisms. This monitoring is performed by counting these microorganisms, which directly measures viable microbial cells, or using other techniques, such as bioluminescence assays, which indirectly detect their presence [10,11]. Although these methods are useful for routine monitoring, they provide a limited view of the whole microbial diversity in the processing environment. To more comprehensively characterize the bacterial diversity in different matrices, researchers have applied metagenomic approaches, which can be defined as DNA analysis of microbial communities in a sample without the need for prior culture [12]. Sequencing of the metagenomic DNA targeting the *16S rRNA* gene is of great interest since it harbors both conserved and variable regions of the bacterial genome [13,14,15,16]. The DNA reads obtained can be compared with reference genetic databases to estimate the taxonomic profile of the environmental samples analyzed [17]. Furthermore, the metataxonomic results can be used to calculate the distribution of microbial populations within a single sample (alpha diversity) and among different samples (beta diversity) [18]. These indexes can help in understanding the microbial ecology of the slaughtering environment because they are supported by more robust data in comparison with traditional methods [19,20].

Despite these advantages, the analysis of metagenomic DNA does not precisely indicate whether the microorganism whose DNA was detected was dead or alive. It has been shown that the presence of an intrinsic microbiota in food industries is not uncommon, with transient and recalcitrant members in the environment [21,22]. However, studies are lacking to show the microbial diversity after pre-operational sanitation in chicken and fish slaughterhouses, especially in Brazil. This information is crucial to improving control strategies to eradicate viable microorganisms that survive SSOPs.

Based on these premises, the objective of the current study was to determine the profile of viable and cultivable microbial communities that survived pre-operational hygienic processes in Brazilian poultry and fish slaughterhouses, combining a culture approach with high-throughput DNA sequencing.

## 2. Materials and Methods

### 2.1. Sampling Procedures

Samplings were conducted in the cutting rooms of two food processing units located in the South and Midwest regions of Brazil. Both slaughterhouses had permission to export their products with official registration by the Brazilian Federal Inspection Service. The fish plant processed mainly *Oreochromis niloticus* (Nile tilapia) with the capacity to slaughter around 150,000 animals/day, and the poultry abattoir slaughtered up to 204,000 animals/day. The animals came from intensive farming systems in the region of the slaughterhouses. Tilapia were raised in excavated ponds, with slaughter conducted between 8 and 9 months of age at a weight of approximately 1 kg. Transport was carried out with the fish alive, in tanks containing potable water and an oxygen supply. Poultry were housed in climate-controlled sheds and slaughtered at approximately 42 days of age, with an average weight of 2.5 kg. Transport was carried out with the animals alive and placed in transport crates.

The temperatures of the cutting rooms were standardized at 12 °C throughout the study. The process of surface sanitization in the cutting rooms consisted of a pre-wash with pressurized water at a temperature of approximately 50 °C. After removing visible residual material, a neutral semi-pasty detergent was applied at a concentration of 3% to 6% using a foam generator. Next, manual scrubbing was performed on surfaces in direct contact with the product, using synthetic fibers and a chlorinated alkaline detergent solution at a concentration of 1% to 4%. After a minimum contact time of 10 min, a rinse with pressurized water at approximately 50 °C was carried out. Finally, a peracetic-acid-based sanitizer was applied at a concentration of 0.1%. Samplings were carried out over 10 consecutive weeks, from June to August 2023.

Samples were collected from food-contact surfaces composed of materials such as polypropylene, polyurethane, and stainless steel, commonly used in poultry and fish slaughterhouses. The points of collection are described in Table 1. These surfaces were selected because they were directly involved in the handling and transportation of food and were included in the SSOPs for routine cleaning and sanitation. In addition, their structural characteristics and the susceptibility to abrasion make them potential sites for the persistence of microorganisms and critical points for hygiene monitoring. To collect samples, surfaces of the poultry (n = 50) and fish (n = 50) slaughterhouses were swabbed with the aid of sterilized molds measuring 10 × 10 cm, totaling 400 cm^2^ for each sample [22,23]. The swabs collected were placed in 10 mL of saline solution composed of 0.85% sodium chloride (w/v), 0.1% bacteriological peptone (w/v), and 0.5% TweenTM 80 (v/v) from Sigma Aldrich (St. Louis, MO, USA) as a neutralizing agent [24].

### 2.2. Sample Processing

The cell suspensions obtained from the swabs for each sampling point (400 cm^2^) were surface-plated (0.1 mL) on Plate Count Agar (PCA, Kasvi, São José dos Pinhais, Paraná, Brazil). After incubation at 36 ± 1 °C for 48 ± 2 h [25], the bacterial community that grew on each plate was harvested by washing with 3 mL of sterilized saline (sodium chloride 0.85 w/v) and transferred by pipette to 5 mL cryotubes to be kept frozen (−20 °C) for further analyses.

### 2.3. Identification of Microbial Diversity Through 16S rRNA Gene Sequencing

Sequencing of 16S rRNA genes for the culturable microbial community from agar plates was performed at the Institute of Biotechnology (IBTEC), São Paulo State University (UNESP), Botucatu Campus, São Paulo, Brazil. Genetic material was extracted using the commercial MagMAX CORE Nucleic Acid Purification Kit (Applied Biosystems™, Waltham, MA, USA) following the manufacturer’s instructions. Purity assessment was performed using the NanoDrop^®^ ND-1000 UV–Vis (Thermo Fisher Scientific, Waltham, MA, USA) equipment. Extracted genetic material was quantified using a Qubit™ 2.0 fluorometer (Invitrogen, Waltham, MA, USA) and the Qubit™ 1X dsDNA High Sensitivity Kit (Invitrogen, Waltham, MA, USA). Considering the manufacturer’s recommendations, samples with 5 ng/µL of genomic DNA were considered for the sequencing step. The V3 and V4 regions of the 16S rRNA gene were used for sequencing on MiSeq equipment (Illumina Inc., San Diego, CA, USA) using the 600-cycle V3 kit. Library preparation consisted of amplification of primers for the V3–V4 region of the 16S rRNA gene (5′-TCGTCGGCAGCGTCAGATGTGTATAAGAGACAGCCTACGGGNGGCWGCAG-3′; 5′-TCTCGTGGGCTCGGAGATGTGTATAAGAGACAGGACTACHVGGGTATCTAATCC-3′; following Klindworth et al. [26] (Illumina Inc., San Diego, CA, USA)). The preparation steps followed the instructions of the 16S Metagenomic Sequencing Library Preparation protocol proposed by Illumina (Illumina Inc., San Diego, CA, USA). Bioinformatics analyses were conducted using QIIME2 software to process sequences and evaluate reads [27]. The DEBLUR program [28] was used to remove lower-quality reads and to categorize the reads into features. For alignment and phylogenetic analysis of the features, MAFFT (Multiple Alignment using Fast Fourier Transform) and FastTree v2.1.11 software were used. The features were taxonomically classified using the SILVA database [29].

### 2.4. Statistical Analysis

The results of 16S rRNA gene sequencing were expressed in operational taxonomic units (OTUs) to calculate the relative frequency (%RF) for phyla, orders, classes, families, genera, and species (whenever possible) for each slaughterhouse. Alpha diversity (Shannon’s diversity index, observed features, Faith’s phylogenetic diversity, and evenness) and beta diversity (Jaccard distance, Bray–Curtis distance, unweighted UniFrac distance, and weighted UniFrac distance) were compared between sample groups of collection points (different processing locations), origin (type of food—poultry or fish), surface (smooth and modular), and material (polypropylene, stainless steel, or polyurethane) using nonparametric tests. Alpha and beta diversity results were expressed graphically using principal coordinates analysis (PCoA). Differences in relative abundances of taxa were assessed by the ANCOM test [30]. A Venn diagram was constructed using Venny 2.1 (https://bioinfogp.cnb.csic.es/tools/venny/, accessed on 4 March 2025) to analyze and describe bacterial genera with RF > 1% for the smooth and modular surfaces.

## 3. Results

### 3.1. Description of Viable and Culturable Microbial Diversity in Poultry and Fish Slaughterhouses After Pre-Operational Sanitation Process

Microbial ecology was determined for 51 samples, 23 (46%) from the poultry slaughterhouse and 28 (56%) from the fish slaughterhouse (Table 2). The excluded samples from the poultry slaughterhouse included 20 (40%) samples with no colony growth on PCA plates and a further 7 (14%) samples, of which 6 had insufficient DNA for sequencing and 1 failed due to poor read quality. In the fish slaughterhouse, 22 (44%) samples were excluded due to the absence of colony growth. The absence of growth on some surfaces may be related to the effectiveness of the sanitization process. Bacterial growth was observed on culture plates regardless of surface type, with microorganisms recovered from both types. A total of 3,047,863 16S rRNA gene sequences were obtained, as follows: 1,817,491 reads from the poultry slaughterhouse (mean = 79,021; minimum = 31,455; maximum = 845,530) and 1,230,372 reads from the fish slaughterhouse (mean = 43,942; minimum = 31,988; maximum = 55,474).

With this approach that combined culture with high-throughput DNA sequencing, all detected taxa at the family, genus, and species levels were reported, with no relative abundance threshold applied. Considering both processing environments, 30 bacterial families were identified. In the poultry slaughterhouse, the highest RF rates comprised *Moraxellaceae* (29.8%), *Microbacteriaceae* (8.1%), *Staphylococcaceae* (7.7%), *Pseudomonadaceae* (5.3%), and *Enterobacteriaceae* (3.6%), as shown in Figure 1. In the fish slaughterhouse, the families *Staphylococcaceae* (18.8%), *Moraxellaceae* (16.9%), *Microbacteriaceae* (14%), Enterobacteriaceae (12.3%), and *Bacillaceae* (8.5%) prevailed (Figure 2).

Regarding genera, 35 were identified in both processing environments. In the poultry slaughterhouse, the most abundant ones were *Acinetobacter* (27.4%), *Staphylococcus* (7.7%), *Pseudomonas* (5.3%), *Aeromonas* (3.1%), and *Streptomyces* (3.1%), as shown in Figure 3. On the other hand, in the fish slaughterhouse, the most frequent genera identified were *Staphylococcus* (27.7%), *Acinetobacter* (17.2%), *Bacillus* (12.5%), *Pseudomonas* (9.6%), and *Enterococcus* (7.1%), as shown in Figure 4.

Due to 16S rRNA gene sequencing limitations, particularly regarding its resolution at the species level, only a few species could be identified. For the poultry slaughterhouse, sequencing analysis of viable and culturable microorganisms revealed the presence of five species, including *Acinetobacter defluvii* (1.5%), *Microvirgula aerodenitrificans* (0.01%), *Sphingobacterium spiritivorum* (0.008%), *Bacteroidetes bacterium* (0.005%), and *Acinetobacter guillouiae* (0.002%). In the fish slaughterhouse, four species were identified, as follows: *Acinetobacter guillouiae* (0.14%), *Pseudomonas geniculata* (0.04%), *Bacteroidetes bacterium* (0.006%), and *Acinetobacter defluvii* (0.005%).

### 3.2. Alpha and Beta Diversity

The microbial diversity of the slaughterhouses was assessed by cultures obtained from swabs, combined with high-throughput DNA sequencing. The results from alpha diversity (richness and uniformity within the sample) and beta diversity (dissimilarity comparing different samples) demonstrated that the type of equipment surface (smooth or modular) was the only factor that influenced alpha diversity, as indicated by the Shannon index (*p* = 0.026) and observed features (*p* = 0.036), shown in Table 3. The point of sampling, the type of material, and the type of food (poultry or fish) did not significantly influence the metrics evaluated.

Regarding beta diversity, the surface type was the only parameter that significantly influenced microbial dissimilarity, as evidenced by the Jaccard distance (*p* = 0.025) and PCoA graphic (shown in Table 3 and Figure 5, respectively).

Although no statistically significant difference was observed (*p* > 0,05) using ANCOM analysis to assess microbial diversity, considering the genera with RF > 1% on modular and smooth surfaces, a Venn diagram was created to better portray the occurrence of the groups of microorganisms observed (Figure 6). The comparative analysis revealed the presence of eight genera (53.3%) in both types of surfaces, while four genera (26.7%) were exclusive to the smooth surface, and three (20%) were exclusive to the modular surface. The *Acinetobacter* genus was the most abundant genus observed on the modular surface, representing 44.3%. On the other hand, on the smooth surface, *Staphylococcus* was the most predominant genus, corresponding to 28% of the observed microorganisms.

## 4. Discussion

It has been shown that the ecology of industrial food processing environments depends on various factors, such as animal microbiota, possible failures in the sanitation processes, the presence of organic residues, and the type of processing surface [19,22,31]. In this sense, it is very important to avoid cross-contamination between products and contact surfaces by implementing environmental monitoring programs [32]. This presumes the use of different techniques to evaluate the efficiency of pre-operational sanitation processes, such as visual inspection, bioluminescence, and microbiological analysis. However, none of these tests offer a detailed description of environmental microbial diversity, which is key information to combat bacterial contaminants.

This study was conducted to provide a clearer picture of the microbial profile found post-sanitation in fish and poultry processing environments. The experimental design proposed started with a culture step before the amplicon-based high-throughput DNA sequencing, which is key to demonstrating the presence of viable and culturable cells that are recalcitrant to the use of sanitizers. Moreover, this combined culture-dependent and culture-independent approach provides an opportunity for viable and culturable cells to produce more signals than the background dead cells [33]. Sequencing of the 16S rRNA gene has been used to evaluate the dynamics of microbial communities in animal product processing environments, as well as to assess factors that interfere with microbial colonization and persistence after pre-operational sanitation [34,35].

To our knowledge, this is the first study in Brazilian slaughterhouses to characterize the microbial diversity of poultry and fish cutting rooms after pre-operational sanitation using a combined approach of culture-dependent and high-throughput culture-independent techniques.

In this study, it was possible to characterize the bacterial diversity in poultry and fish slaughterhouses by sequencing the totality of *16S rRNA* genes from cultures obtained from 51 samples, showing that both alpha and beta diversities were influenced by the type of surface (Table 3). Alpha diversity, assessed by the Shannon index (*p* = 0.026) and observed features (*p* = 0.036), revealed significant differences in both richness and uniformity within the samples. Furthermore, beta diversity was essential for comparing the microbial communities between the two industries, with the Jaccard distance showing a significant difference (*p* = 0.025) [18]. These results suggest bacterial colonization and survival were possibly influenced by the presence of irregular and recessed surfaces (such as in modular equipment), which may favor bacterial adherence and hinder the homogeneous dispersion of sanitizing agents, favoring biofilm formation [36]. Similarly, Tadielo et al. [37] also reported that modular-type polypropylene conveyors exhibited significantly greater microbial contamination than smooth-type polystyrene conveyors before the cleaning process, reinforcing that surface characteristics play a crucial role in microbial retention and contamination dynamics in food processing environments.

Moreover, the results of the present study revealed that the sample source did not significantly influence diversity indexes, and three genera were highly prevalent in the poultry and fish industry, namely *Acinetobacter*, *Staphylococcus*, and *Pseudomonas* (Figure 3 and Figure 4).

These results are consistent with those from Tadielo et al. [37], who reported increased RF of *Acinetobacter* sp. and *Pseudomonas* sp. in a chicken slaughterhouse after the pre-operational hygiene process in Brazil. These findings are corroborated by Sinlapapanya [38], who also found *Acinetobacter* sp. and *Pseudomonas* sp. after the cleaning and sanitation in a fish processing establishment located in Thailand. According to Møretrø & Langsrud [21], *Acinetobacter* sp. and *Pseudomonas* sp. are usually found in the same niches in food processing plants because they present similar growth and survival characteristics, with the ability to tolerate harsh conditions.

The *Acinetobacter* genus is commonly found both in chicken and fish meat [16,39], as well as on industrial surfaces used for food handling [40,41]. Its environmental prevalence may be related to the contamination of raw materials in contact with soil, water, and vegetation [42]. Moreover, it may tolerate low temperature, low availability of nutrients, cleaners/disinfectants, and shear forces [21], which may favor biofilm formation [38,43]. It is interesting to note that *Acinetobacter* sp. can dominate multispecies biofilms in vitro [44], which may partially explain its high prevalence (44.3%) on the surface of modular equipment observed in the present study. In addition, *Acinetobacter* spp. has been recognized for its ability to acquire and disseminate antimicrobial resistance genes, enabling the emergence of microorganisms with a multidrug-resistant phenotype [45,46]. This characteristic, combined with the potential for biofilm formation and the ability to persist in food processing environments, represents significant concerns for food safety, since it can contribute to the transmission of resistant or multidrug-resistant strains through cross-contamination along the food processing chain [47].

*Pseudomonas* spp. are psychrotrophic bacteria with a high ability to adapt to diverse environments, which can be attributed partially to the presence of many regulatory genes in their genome [48]. The ability of biofilm formation by *Pseudomonas* sp. is very important in food processing environments because it not only protects the bacterium against sanitizers, but this species has also been regarded as a primary colonizer that facilitates the adhesion of other bacteria [49,50,51]. This genus is recognized as a member of the natural microbiota of poultry [52] and fish [53], which helps to explain the high prevalence observed in this study. Other authors have reported that Pseudomonas sp. is a major contaminant in environments where the processing of animal products takes place [22,54].

On the other hand, in this study, the highest prevalence of *Staphylococcus* spp. was observed in the fish industry (27.7%). Considering that staphylococci are commonly found in skin and mucous tissues of humans and animals, the presence of this microorganism in food processing environments may suggest inadequate food handling practices [55]. Besler & Kılınç [56] evaluated the effect of various sanitizers on *Staphylococcus* sp. isolates from the fish slaughter industry, reporting that lower concentrations of the antimicrobial compounds tested did not present bactericidal action. In addition, those authors reported that staphylococci were highly prevalent (28%) on smooth surfaces (e.g., stainless steel), which may indicate their strong ability to form biofilms [57].

In the poultry abattoir investigated in this study, *Aeromonas* (3.1%) and *Streptomyces* (3.1%) were also highly abundant, which is consistent with their widespread presence as part of the microbiota of these animals [58,59]. *Aeromonas* sp. is a strong biofilm former [60], besides being tolerant of sanitizers in chicken slaughtering plants [37].

In the fish processing industry evaluated in this research, *Bacillus* (12.5%) and *Enterococcus* (7.1%) were also among the most prevalent genera. *Bacillus* sp. has a high ability to adapt to diverse environments, besides being able to form spores, presenting tolerance to very stressful conditions [38,61]. Concerning enterococci, Mendoza et al. [62] reported a high prevalence in several spots of a production chain of Nile tilapia, besides their presence in the water and healthy animals. The presence of enterococci in the fish processing chain is possibly an indicator of water contamination.

Considering that metagenomic DNA analysis does not distinguish between live and dead microorganisms, prior cultivation becomes a methodology that enables the exclusive evaluation of viable and culturable cells present in the samples. The use of prior cultivation followed by 16S rRNA gene sequencing has been employed to characterize the microbial diversity of food processing facilities [43,63]; milk, dairy products, and cheese [63,64]; plant-based products [65]; human milk [66]; as well as in the characterization of the microbiota of humans [67,68] and livestock animals [69]. However, this approach does not account for viable but non-culturable (VBNC) microorganisms, which also represent a challenge in assessing the effectiveness of sanitation procedures. These organisms remain metabolically active but lose the ability to grow on conventional culture media under laboratory conditions, making their detection difficult [70]. Nevertheless, the strategy adopted in our study focuses on the fraction of microorganisms with the greatest practical relevance to the food industry. These viable microorganisms are particularly important for evaluating the risk of cross-contamination and biofilm formation, both critical aspects in the control of industrial processes.

Peracetic acid is commonly used in the food industry because of its ability to inactivate microorganisms [71,72]. However, some bacteria may persist after the sanitization process, since several factors can compromise its effectiveness, including the concentration, time, and temperature of application [73], biofilm formation [74], and microbial interactions [51].

Therefore, the recovery of these viable isolates enables future studies aimed at identifying phenotypic and genetic factors associated with tolerance to sanitation programs, thereby enhancing the understanding of the mechanisms that favor microbial persistence in industrial environments and supporting the development of more effective and targeted control measures for food processing facilities [75,76].

In broad terms, the results of this study contributed to a better understanding of the resident microbial diversity of poultry and fish processing environments after pre-operation sanitation procedures. It was shown that the characteristics of the surfaces influenced the extent of microbial colonization, challenging the control strategies applied to control microbial contamination in different spots. The results also indicate the need for studies on microbial communities to evaluate the effectiveness of sanitizers and to develop novel strategies to combat microbial contamination in food processing plants.

## 5. Conclusions

This study revealed the presence of viable and culturable microbial diversity after the cleaning process, highlighting that surface characteristics influence the diversity observed. *Acinetobacter* was predominant on modular surfaces, while *Staphylococcus* prevailed on smooth surfaces. These results suggest that the type of equipment surface plays a critical role in shaping microbial communities in the food industry. Understanding this influence can support the development of more effective hygiene strategies, reinforcing food quality and safety.

## Figures and Tables

**Figure 1 foods-14-02387-f001:**
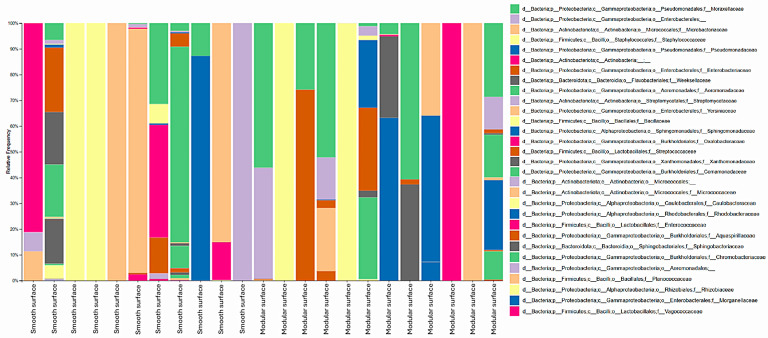
Diversity of bacterial families identified on smooth and modular surfaces after the pre-operational sanitation process in a Brazilian poultry slaughterhouse. Swabs from the surfaces were cultured on Plate Count Agar, total DNA was extracted from the grown microbial communities, and the V3 and V4 regions of 16S rRNA genes were amplified and sequenced with high-throughput technology. All identified bacterial families are shown, with no abundance threshold applied.

**Figure 2 foods-14-02387-f002:**
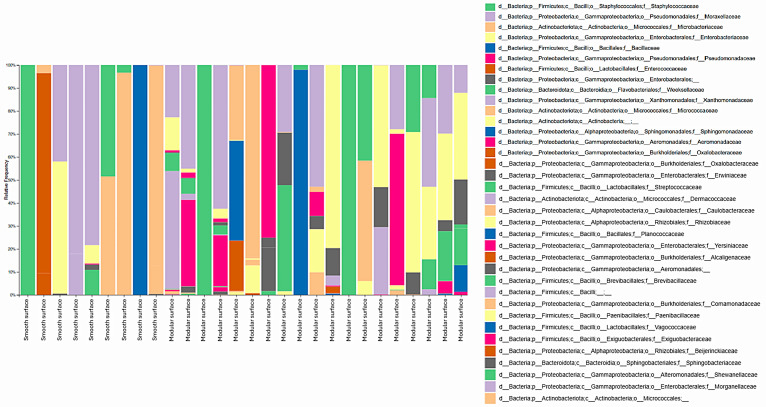
Diversity of bacterial families identified on smooth and modular surfaces after the pre-operational sanitation process in a Brazilian fish slaughterhouse. Swabs from the surfaces were cultured on Plate Count Agar, total DNA was extracted from the grown microbial communities, and the V3 and V4 regions of 16S rRNA genes were amplified and sequenced with high-throughput technology. All identified bacterial families are shown, with no abundance threshold applied.

**Figure 3 foods-14-02387-f003:**
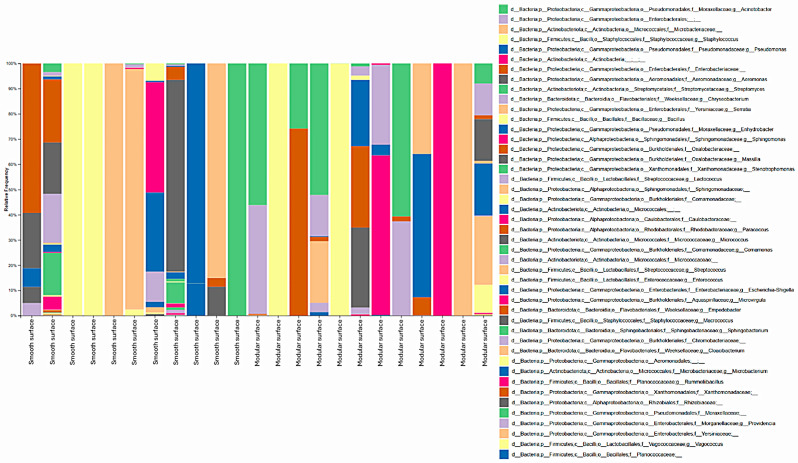
Diversity of bacterial genera identified on smooth and modular surfaces after the pre-operational sanitation process in a Brazilian poultry slaughterhouse. Swabs from the surfaces were cultured on Plate Count Agar, total DNA was extracted from the grown microbial communities, and the V3 and V4 regions of 16S rRNA genes were amplified and sequenced with high-throughput technology. All identified bacterial genera are shown, with no abundance threshold applied.

**Figure 4 foods-14-02387-f004:**
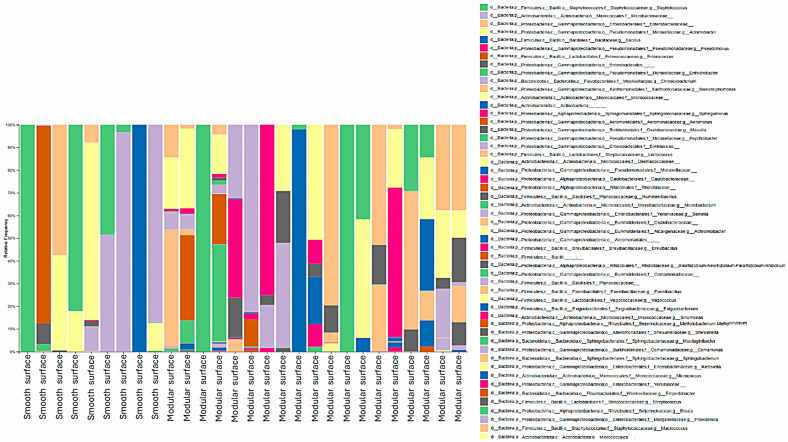
Diversity of bacterial genera identified on smooth and modular surfaces after the pre-operational sanitation process in a Brazilian fish slaughterhouse. Swabs from the surfaces were cultured on Plate Count Agar, total DNA was extracted from the grown microbial communities, and the V3 and V4 regions of 16S rRNA genes were amplified and sequenced with high-throughput technology. All identified bacterial genera are shown, with no abundance threshold applied.

**Figure 5 foods-14-02387-f005:**
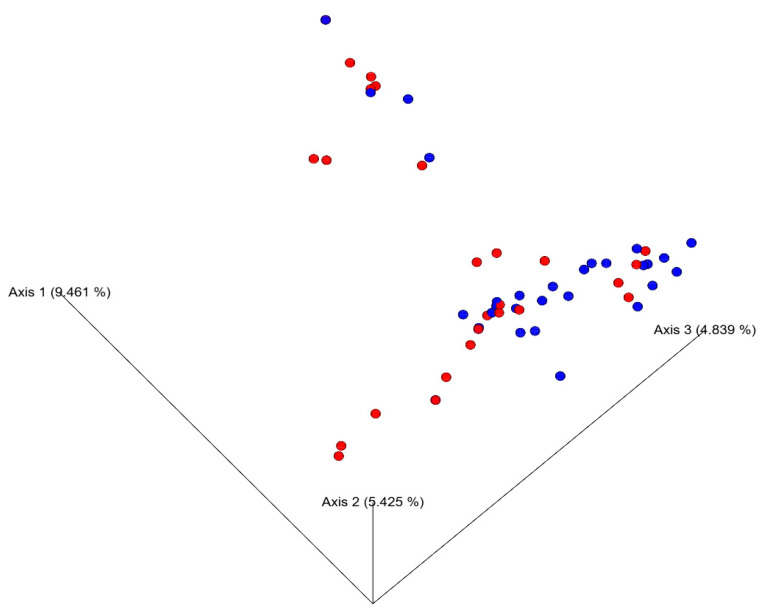
Principal coordinates analysis (PCoA) based on Jaccard dissimilarity of bacterial communities from modular (blue) and smooth (red) surfaces after pre-operational sanitation in Brazilian poultry and fish slaughterhouses. Axes 1, 2, and 3 explain 9.461%, 5.425%, and 4.839% of the variation, respectively. The distribution of samples shows a partial separation by surface type, suggesting that surface characteristics may influence bacterial composition after cleaning.

**Figure 6 foods-14-02387-f006:**
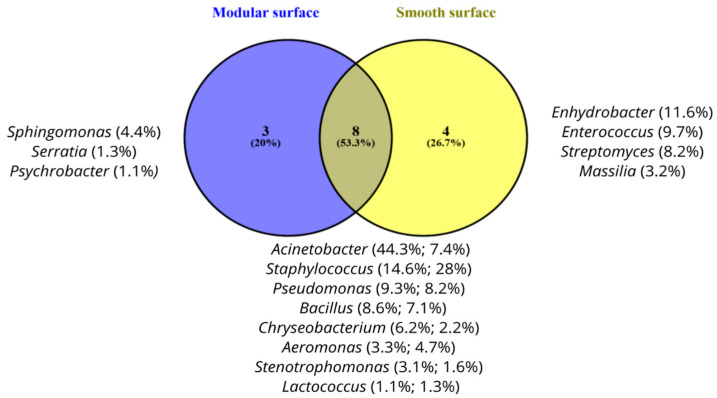
Venn diagram based on bacterial genera with relative frequency (RF) > 1% identified after pre-operational sanitation on modular and smooth surfaces, respectively, from Brazilian fish and poultry slaughterhouses.

**Table 1 foods-14-02387-t001:** Location and quantity of samples evaluated on different equipment surfaces in Brazilian poultry and fish slaughterhouses.

Collection Points	N	%
Poultry slaughterhouse (n = 50)		
Modular polypropylene conveyor belt	10	20
Modular polypropylene conveyor belt	10	20
Stainless steel table	10	20
Smooth polyurethane conveyor belt	10	20
Modular polypropylene conveyor belt	10	20
Fish slaughterhouse (n = 50)		
Modular polypropylene conveyor belt	10	20
Smooth polypropylene cutting board	10	20
Skin machine (stainless steel)	10	20
Modular polypropylene conveyor belt	10	20
Modular polypropylene conveyor belt	10	20

**Table 2 foods-14-02387-t002:** Presence and absence of bacterial growth on PCA plates from samples evaluated on equipment surfaces in Brazilian poultry and fish slaughterhouses.

Collection Points	N +	%	N −	%
Poultry slaughterhouse (n = 50)				
Modular polypropylene conveyor belt	4	8	6	12
Modular polypropylene conveyor belt	7	14	3	6
Stainless steel table	6	12	4	8
Smooth polyurethane conveyor belt	7	14	3	6
Modular polypropylene conveyor belt	6	12	4	8
Fish slaughterhouse (n = 50)				
Modular polypropylene conveyor belt	6	12	4	8
Smooth polypropylene cutting board	5	10	5	10
Skin machine (stainless steel)	4	8	6	12
Modular polypropylene conveyor belt	5	10	5	10
Modular polypropylene conveyor belt	8	16	2	4

(+) indicates the number and percentage of samples with bacterial growth on PCA plates. (−) indicates the number and percentage of samples with no bacterial growth on PCA plates.

**Table 3 foods-14-02387-t003:** Alpha and beta diversity parameters of culturable microbial communities observed on surfaces of poultry and fish slaughterhouses after pre-operational sanitation, considering different parameters (type of meat product, collection site, type of surface, and surface material).

Categories	Alpha Diversity *				Beta Diversity **			
	Shannon’s Diversity Index	Observed Features	Faith’s Phylogenetic Diversity	Evenness	Jaccard Distance	Bray–Curtis distance	Unweighted UniFrac Distance	Weighted UniFrac Distance
Type of food	0.283	0.311	0.194	0.732	0.067	0.109	0.122	0.621
Sampling point	0.304	0.369	0.610	0.504	–	–	–	–
Type of surface	0.026	0.036	0.110	0.145	0.025	0.082	0.227	0.215
Type of material	0.364	0.584	0.455	0.204	0.759	0.721	0.760	0.950

* Kruskal–Wallis test, with statistical significance defined by *p* < 0.05. ** Pseudo-F test, with statistical significance defined by *p* < 0.05.

## Data Availability

The original contributions presented in the study are included in the article. Further inquiries can be directed to the corresponding author.

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
