# Peer review of "Total Culturable Microbial Diversity of Food Contact Surfaces in Poultry and Fish Processing Industries After the Pre-Operational Cleaning Process"

_foods, 2025, doi:10.3390/foods14132387_

Round 1

Reviewer 1 Report

Comments and Suggestions for Authors

Overall Comments:

This study investigates the culturable microbial diversity on food-contact surfaces in Brazilian poultry and fish processing plants after pre-operational cleaning, combining culture-based methods with high-throughput DNA sequencing. The research design is sound, the analytical methods are advanced, and the findings have practical implications for hygiene control in the food industry. However, the manuscript requires improvements in methodological descriptions, presentation of results, depth of discussion, and language clarity. A revised version would be suitable for reconsideration for publication.

Specific Comments:

(1)In "2.1 Sampling procedures," the description of the sanitation process mentions using "neutral semi-pasty detergent and chlorinated alkaline liquid detergent" but does not specify contact time, temperature, or concentrations (other than 0.1% peracetic acid). These details are critical for assessing the effectiveness of cleaning steps and should be added.

(2)In "3.1," the authors note that some samples were excluded due to failed culturing, insufficient DNA, or bioinformatics issues, but they do not report the exact number of excluded samples or how this affected the statistical power. Clarifying the sample size reduction (e.g., from 100 to 51) and its impact is essential for transparency.

(3)The "ANCOM test" (Lines 188–189) should specify parameters (e.g., p-value threshold).

(4)The discussion on Acinetobacter spp. (Lines 270–271) should address potential food safety risks (e.g., antibiotic resistance).

(5)The inability of culture methods to detect viable but non-culturable (VBNC) microorganisms should be acknowledged (expand Lines 320–323).

(6)Ensure consistent formatting for DOIs and publication details across all citations (e.g., some entries include DOIs, others do not). Follow the journal’s reference style uniformly.

(7) Some sections, such as the introduction to SSOPs (Paragraphs 15–16), contain repetitive descriptions. Streamline the text to improve readability.

Author Response

Response to Reviewer #1

(1)In "2.1 Sampling procedures," the description of the sanitation process mentions using "neutral semi-pasty detergent and chlorinated alkaline liquid detergent" but does not specify contact time, temperature, or concentrations (other than 0.1% peracetic acid). These details are critical for assessing the effectiveness of cleaning steps and should be added.

Based on the suggestions, a new piece of information was inserted in the manuscript: “The process of surface sanitization in the cutting rooms consisted of a pre-wash with pressurized water at a temperature of approximately 50 °C. After removing visible residual material, a neutral semi-pasty detergent was applied at a concentration of 3% to 6%, using a foam generator. Next, manual scrubbing was performed on surfaces in direct contact with the product, using synthetic fibers and a chlorinated alkaline detergent solution at a concentration of 1% to 4%. After a minimum contact time of 10 minutes, a rinse with pressurized water at approximately 50 °C was carried out. Finally, a peracetic acid-based sanitizer was applied at a concentration of 0.1%.” (lines 101-108)

2)In "3.1," the authors note that some samples were excluded due to failed culturing, insufficient DNA, or bioinformatics issues, but they do not report the exact number of excluded samples or how this affected the statistical power. Clarifying the sample size reduction (e.g., from 100 to 51) and its impact is essential for transparency.

Based on the suggestions, the manuscript was modified as follows: “The excluded samples from the poultry slaughterhouse included 18 (36%) samples with no colony growth on PCA plates and 9 (18%) samples with insufficient DNA for sequencing or that failed to meet the quality criteria during bioinformatics analysis. In the fish slaughterhouse, 22 (44%) samples were excluded due to the absence of colony growth.  Bacterial growth on culture plates occurred regardless of the type of surface, with micro-organisms being recovered from both surface types.” (lines 173-178)

(3)The "ANCOM test" (Lines 188–189) should specify parameters (e.g., p-value threshold).

To clarify this point, a phrase was included in the manuscript: “Although no statistically significant difference was observed (p > 0,05) using ANCOM analysis to assess microbial diversity, considering the genera with RF > 1% on modular and smooth surfaces, a Venn diagram was created to better portray the occurrence of the groups of microorganisms observed (Figure 6). (line 224)

(4)The discussion on Acinetobacter spp. (Lines 270–271) should address potential food safety risks (e.g., antibiotic resistance).

The manuscript was modified as suggested: “In addition, Acinetobacter spp. has been recognized for its ability to acquire and disseminate antimicrobial resistance genes, enabling the emergence of microorganisms with a multidrug-resistant phenotype [45,46]. This characteristic, combined with the potential for biofilm formation and the ability to persist in food processing environments, represents significant concerns for food safety, since it can contribute to the transmission of resistant or multidrug-resistant strains through cross-contamination along the food processing chain [47].” (lines 319-325)

(5)The inability of culture methods to detect viable but non-culturable (VBNC) microorganisms should be acknowledged (expand Lines 320–323).

This observation was included in the manuscript: “However, this approach does not account for viable but non-culturable (VBNC) microorganisms, which also represent a challenge in assessing the effectiveness of sanitation procedures. These organisms remain metabolically active but lose the ability to grow on conventional culture media under laboratory conditions, making their detection difficult [70].” (lines 361-365)

(6)Ensure consistent formatting for DOIs and publication details across all citations (e.g., some entries include DOIs, others do not). Follow the journal’s reference style uniformly.

References have been corrected accordingly.

(7) Some sections, such as the introduction to SSOPs (Paragraphs 15–16), contain repetitive descriptions. Streamline the text to improve readability.

We appreciate your suggestion and changes have been made throughout the text.

Reviewer 2 Report

Comments and Suggestions for Authors

General comments on the Introduction: this is not of sufficient length to convey the complexity of the landscape of slaughterhouse food safety regulations, practices, and prerequisite programs. The authors should utilize and discuss existing research studies alongside their most important statements for establishing a narrative for their own work (e.g., in lines 53-55, the phrase "which has been proven to be a very useful approach" does not provide detail for what the approach is, why it is considered useful, how it is considered useful, and why it is relevant to the work of this manuscript).

General comments on the Materials and Methods: Please explain the materials that samples were collected from and why they are relevant (i.e., how they are used in slaughterhouses and what relationship they have to the prerequisite programs identified in the Introduction section). How were "cultivable" isolates identified/obtained, as mentioned in the objective statement?

General comments on the Results: What abundance threshold was set for reporting (this will matter when discussing the significance of only five and four identified species in poultry and fish slaughterhouses, respectively).

Figures 1-4: Please reference the abundance threshold for inclusion in this figure in the figure title.

Figure 6: What is the significance of comparing the overlap of general across both poultry and fish slaughterhouses together?

Author Response

Response to Reviewer #2

General comments on the Introduction: this is not of sufficient length to convey the complexity of the landscape of slaughterhouse food safety regulations, practices, and prerequisite programs. The authors should utilize and discuss existing research studies alongside their most important statements for establishing a narrative for their own work (e.g., in lines 53-55, the phrase "which has been proven to be a very useful approach" does not provide detail for what the approach is, why it is considered useful, how it is considered useful, and why it is relevant to the work of this manuscript).

Based on the suggestions, a new piece of information was inserted in the manuscript:

“To this end, Food Safety Management Systems (FSMS) are applied, which are a set of processes, procedures, and tools, based on standards and regulations, developed to control hazards and ensure compliance with food safety requirements. These systems cover, among others, prerequisite programs such as Good Hygiene Practices (GHP) and Sanitation Standard Operating Procedures (SSOP) [4,5]. (lines 46-50)

“Industries usually assess the effectiveness of SSOPs by monitoring indicator microorganisms. This monitoring is performed by counting these microorganisms, which directly measures viable microbial cells, or using other techniques, such as bioluminescence as-says, which indirectly detect their presence [10,11]. Although these methods are useful for routine monitoring, they provide a limited view of the whole microbial diversity in the processing environment.” (lines 58-63)

General comments on the Materials and Methods: Please explain the materials that samples were collected from and why they are relevant (i.e., how they are used in slaughterhouses and what relationship they have to the prerequisite programs identified in the Introduction section). How were "cultivable" isolates identified/obtained, as mentioned in the objective statement?

We modified the manuscript and introduced paragraphs to clarify the points raised by the reviewer: “Samples were collected from food-contact surfaces composed of materials such as polypropylene, polyurethane, and stainless steel, commonly used in poultry and fish slaughterhouses. The points of collection are described in Table 1. These surfaces were selected because they were directly involved in the handling and transportation of food, and were included in the SSOPs for routine cleaning and sanitation. In addition, their structural characteristics and the susceptibility to abrasion make them potential sites for the persistence of microorganisms and critical points for hygiene monitoring. (lines 110-116)

The culturable isolates were identified/obtained as described in sections 2.2 and 2.3 of the manuscript. They refer to microbial communities that were able to grow on Plate Count Agar (PCA) after surface swabbing. In short, after incubation, total DNA was extracted from the resulting microbial growth, followed by amplification and sequencing of the V3-V4 regions of the 16S rRNA gene, allowing characterization of the viable and cultivable microbiota.

General comments on the Results: What abundance threshold was set for reporting (this will matter when discussing the significance of only five and four identified species in poultry and fish slaughterhouses, respectively).

We have clarified in the manuscript that all detected taxa at the family, genus, and species levels were reported, with no relative abundance threshold was applied. This information was included in the Results section before presenting the taxonomic profiles.

“With this approach that combined culture with high-throughput DNA sequencing, all detected taxa at the family, genus, and species levels were reported, with no relative abundance threshold applied.   (lines 183-185)

Figures 1-4: Please reference the abundance threshold for inclusion in this figure in the figure title.

We have updated the figure titles for Figures 1–4 to clarify that no relative abundance threshold was applied for inclusion; all detected taxa are reported in the figures.

Figure 6: What is the significance of comparing the overlap of general across both poultry and fish slaughterhouses together?

The comparison of the overlapping genera across both poultry and fish slaughterhouses is significant because there is limited data available in Brazil regarding the shared microbial communities in these processing environments. Moreover, the regions where this study was conducted are nationally recognized for their strong poultry and aquaculture industries, which are some of the most important sectors in Brazilian agribusiness. Therefore, our findings represent robust and meaningful data that contribute to improving food safety practices within these important segments of the national food supply chain.

Reviewer 3 Report

Comments and Suggestions for Authors

Dear authors,

 I am sending you some of my suggestions:

  1. Explain in more detail the surface washing procedure, the composition of the detergent and disinfectant, the time period and the frequency and method of washing (used equipment), the temperature of the water and washing liquid.
  2. Describe the origin, method of growing and transporting fish to the slaughterhouse.
  3. Describe the origin, method of keeping and transporting poultry to the slaughterhouse.
  4. Additionally explain the sources of contamination of the working environment in the slaughterhouse with identified microorganisms, the conditions that these bacteria require and their hygienic significance?
  5. Specify the method and reference according to which the sampling was carried out, i.e. taking swabs and preparing the composition of the medium in which the swabs were placed.
  6. List the results of microbiological testing of the water used in the slaughterhouse, i.e. washing.
  7. Describe DNA extraction methods, evaluation of DNA purity, quantification of purified material, sequencing procedure, name and authors of methods, protocol steps, amplification procedure. Specify the method according to which the sequencing was done.
  8. Explain why Tween TM 80 was used as an addition to the nutrient medium and why in a concentration of 0.1%? According to which reference?
  9. Explain why a temperature of 36 +- 1 °C was used for incubation? Indicate the method and reference according to which the microbiological test was performed and the selection of the nutrient medium.
  10. Explain how the type of work surface affects the type and number of isolated bacteria?
  11. State the primer manufacturer, method description and reference
  12. To my knowledge, metagenomics is used to identify bacteria through genetic material directly from a sample, not from bacterial cultures. The question arises of the influence of this method on the obtained results. What are the limitations and disadvantages of this method?
  13. Explain in more detail the principles of the bioinformatics and statistical methods used and provide references where they are missing.
  14. A very large number of bacteriologically negative samples were determined - almost half. Explain in more detail how many samples could not be tested due to lack of bacterial growth on the plate, lack of DNA, or failure to meet bioanalysis criteria? (which bioanalysis criteria were not met?). Explain the causes of these defects. From which working surfaces were the negative samples obtained?
  15. Explain the difference in results between the obtained genera and species of bacteria, no match?
  16. Bacteroidetes bacterium - explain what species it is? I didn't find it in the systematics.
  17. Explain the difference in the attachment of Staphylococcus (is it S. aureus?) and Acinetobacter to smooth and flat surfaces, because both create biofilms
  18. Check the DOI number of reference 6, as it shows that it is non-existent
  19. Check whether references 13, 25, 29, 30, 31, 35, 46, 50 are listed in the right place in the text, whether they correspond to the specified text. I think there are discrepancies, so it should be supplemented with more specific references.
  20. I think the smallest text on the pictures should be clearer
  21. For reference 26, only the site is listed, it should be supplemented with the full name
  22. I suggest that the authors mark the changes and additions in the text with some color, so that it is more noticeable for reading.

If you have any questions or concerns, I am at your disposal.

Sincerely,

Author Response

Response to Reviewer #3

 Explain in more detail the surface washing procedure, the composition of the detergent and disinfectant, the time period and the frequency and method of washing (used equipment), the temperature of the water and washing liquid.

The explanation requested was inserted in the manuscript: “The process of surface sanitization in the cutting rooms consisted of a pre-wash with pressurized water at a temperature of approximately 50 °C. After removing visible residual material, a neutral semi-pasty detergent was applied at a concentration of 3% to 6%, using a foam generator. Next, manual scrubbing was performed on surfaces in direct contact with the product, using synthetic fibers and a chlorinated alkaline detergent solution at a concentration of 1% to 4%. After a minimum contact time of 10 minutes, a rinse with pressurized water at approximately 50 °C was carried out. Finally, a peracetic acid-based sanitizer was applied at a concentration of 0.1%.” (lines 101-108)

Describe the origin, method of growing and transporting fish to the slaughterhouse. Describe the origin, method of keeping and transporting poultry to the slaughterhouse.

The descriptions requested were included in the manuscript: “The animals came from intensive farming systems in the region of the slaughterhouses. Tilapia were raised in excavated ponds, with slaughter conducted between 8 and 9 months of age and weighing approximately 1 kg. Transport was done with the fish alive, in tanks containing potable water and an oxygen supply. Poultry were housed in climate-controlled sheds and slaughtered at approximately 42 days of age, with an average weight of 2.5 kg. Transport was carried out with the animals alive and placed in transport crates.” (lines 93-99)

Additionally explain the sources of contamination of the working environment in the slaughterhouse with identified microorganisms, the conditions that these bacteria require and their hygienic significance?

We discussed the sources of contamination in the slaughterhouse environment, the conditions required by the identified microorganisms, and their hygienic significance throughout the discussion section, especially for each microorganism. Since this information is already present in the manuscript during the discussion of the main microorganisms found, no further changes were made in response to this comment.

Specify the method and reference according to which the sampling was carried out, i.e. taking swabs and preparing the composition of the medium in which the swabs were placed. Explain why Tween TM 80 was used as an addition to the nutrient medium and why in a concentration of 0.1%? According to which reference?

We have inserted references specifying the method according to which the sampling was carried out, including the procedures for taking swabs and preparing the composition of the medium in which the swabs were placed. (line 121)

List the results of microbiological testing of the water used in the slaughterhouse, i.e. washing.

Unfortunately, the slaughterhouses did not provide official data regarding the microbiological quality of the water used in their operations. However, according to Brazilian legislation, companies are required to ensure and attest to the quality and safety of the processing water used in various industrial operations. A limitation of our study was that we were unable to collect and analyze water samples during the research period.

Describe DNA extraction methods, evaluation of DNA purity, quantification of purified material, sequencing procedure, name and authors of methods, protocol steps, amplification procedure. Specify the method according to which the sequencing was done. State the primer manufacturer, method description and reference.

These methodological details are described in section 2.3 of the methodology. Based on the suggestions, new information was inserted in the manuscript: “Considering the manufacturer's recommendations, samples with 5 ng/µL of genomic DNA were considered for the sequencing step.” (lines 139-141)

Explain why a temperature of 36 +- 1 °C was used for incubation? Indicate the method and reference according to which the microbiological test was performed and the selection of the nutrient medium.

We opted to evaluate the total mesophilic microorganisms because some psychrotrophic microorganisms are also able to grow at this incubation temperature (36 ± 1 °C). The aim of the study was not to compare different incubation temperatures for the selection of this microbiota, but rather to use the official method, which also establishes this incubation temperature. The method and reference used for the microbiological test were inserted in section 2.2.

Explain how the type of work surface affects the type and number of isolated bacteria?

A text was included to clarify this point: “These results suggest bacterial colonization and survival were possibly influenced by the presence of irregular and recessed surfaces (such as in modular equipment), which may favor bacterial adherence and hinder the homogeneous dispersion of sanitizing agents, favoring biofilm formation [36]. Similarly, Tadielo et al. [37] also reported that modular-type polypropylene conveyors exhibited significantly greater microbial contamination than smooth-type polystyrene conveyors before the cleaning process, reinforcing that surface characteristics play a crucial role in microbial retention and contamination dynamics in food processing environments.” (lines 292-299)

To my knowledge, metagenomics is used to identify bacteria through genetic material directly from a sample, not from bacterial cultures. The question arises of the influence of this method on the obtained results. What are the limitations and disadvantages of this method?

We have expanded the discussion regarding the limitations of the technique used, in order to further clarify this aspect. (lines 374-378)

Explain in more detail the principles of the bioinformatics and statistical methods used and provide references where they are missing.

Explanation: Alpha diversity, assessed by the Shannon index (p = 0.026) and Observed Features (p = 0.036), revealed significant differences in both richness and uniformity within the samples. Furthermore, beta diversity was essential for comparing the microbial communities between the two industries, with the Jaccard distance showing a significant difference (p = 0.025) [18]. (lines 288-292)

A very large number of bacteriologically negative samples were determined - almost half. Explain in more detail how many samples could not be tested due to lack of bacterial growth on the plate, lack of DNA, or failure to meet bioanalysis criteria? (which bioanalysis criteria were not met?). Explain the causes of these defects. From which working surfaces were the negative samples obtained?

This point was clarified in the manuscript, as follows: “The excluded samples from the poultry slaughterhouse included 18 (36%) samples with no colony growth on PCA plates and 9 (18%) samples with insufficient DNA for sequencing or that failed to meet the quality criteria during bioinformatics analysis. In the fish slaughterhouse, 22 (44%) samples were excluded due to the absence of colony growth.  Bacterial growth on culture plates occurred regardless of the type of surface, with micro-organisms being recovered from both surface types.” (lines 173-178)

Information on methodological criteria regarding the extraction, purification, and sequencing of isolates can be found in the section 2.3.

Explain the difference in results between the obtained genera and species of bacteria, no match?

Explanation included in the manuscript: “Due to 16S rRNA gene sequencing limitations, particularly regarding its resolution at the species level, only a few species could be identified.” (lines 197-198)

Bacteroidetes bacterium - explain what species it is? I didn't find it in the systematics.

In our study, it was not a priority to identify the most prevalent taxa at the species level, but rather to analyze the distribution of the main groups of microorganisms identified in each environment, which can directly impact the effectiveness of sanitation programs. Additionally, due to the limitations of the technique in distinguishing taxa at the species level, we chose to present the taxonomic classification primarily at the genus level for better data visualization and interpretation.

Explain the difference in the attachment of Staphylococcus (is it S. aureus?) and Acinetobacter to smooth and flat surfaces, because both create biofilms.

Both genera are capable of forming biofilms. However, their predominance on different surfaces may reflect distinct ecological niches and contamination routes, rather than just variations in adhesion mechanisms. As reported in our study, Staphylococcus can often be associated with human contamination during food handling and processing, while Acinetobacter is more related to contamination of raw materials through contact with soil, water, and vegetation.

However, regardless of the type of surface, growth temperature, or the presence of a substrate, both microorganisms play a relevant role in biofilm formation and in tolerance to industrial sanitation programs, as reported in several studies [21,38-44,54,55].

Check the DOI number of reference 6, as it shows that it is non-existent

The DOI number of reference 6 has been corrected. Thank you for the observation.

Check whether references 13, 25, 29, 30, 31, 35, 46, 50 are listed in the right place in the text, whether they correspond to the specified text. I think there are discrepancies, so it should be supplemented with more specific references.

We have checked the references. Thank you for the careful review of our manuscript.

I think the smallest text on the pictures should be clearer

Due to software limitations, it is not possible to adjust the clarity of the smallest text in the figures.

For reference 26, only the site is listed, it should be supplemented with the full name

The reference has been corrected.

Round 2

Reviewer 3 Report

Comments and Suggestions for Authors

Dear authors, I am sending you a few more observations of mine:

  1.  It would be useful to supplement the text with information on the sensitivity of bacteria to a 0.1% solution of peracetic acid - to try to explain survival of bacteria after washing and disinfecting surfaces
  2. I would say that reference 24. Listed in the chapter Materials and methods regarding the application of Tween 80 is not appropriate. Support it with a more specific reference.
  3. The discussion in the text does not mention figure 1 nor is it discussed in the text.
  4. Explain figure 3, the meaning of axis 1, 2 and 3.
  5. It would be good to further specify how many samples were not tested due to lack of DNA, and how many due to failure to meet the bioanalysis criteria? Which bioanalysis criteria were not met?.
  6. It would be useful to list which samples - from which surfaces - were bacteriologically negative after incubation and to try to explain. 
  7. I did not notice that the sequencing results were presented through diagrams and figures for the five identified bacterial species mentioned in the paper.

Sincerely,

Author Response

  1. It would be useful to supplement the text with information on the sensitivity of bacteria to a 0.1% solution of peracetic acid - to try to explain survival of bacteria after washing and disinfecting surfaces

Based on the suggestions, the manuscript was modified (lines 376-380): “Peracetic acid is commonly used in the food industry because of its ability to inactivate microorganisms [71,72]. However, some bacteria may persist after the sanitization process, since several factors can compromise its effectiveness, including the concentration, time, and temperature of application [73], biofilm formation [74], and microbial interactions [75]”.

  1. I would say that reference 24. Listed in the chapter Materials and methods regarding the application of Tween 80 is not appropriate. Support it with a more specific reference.

Based on the suggestion, a new reference was inserted.

  1. The discussion in the text does not mention figure 1 nor is it discussed in the text.

Figure 1 is mentioned in the Results section.

“In the poultry slaughterhouse, the highest RF rates comprised Moraxellaceae (29.8%), Microbacteriaceae (8.1%), Staphylococcaceae (7.7%), Pseudomonadaceae (5.3%), and Enterobacteriaceae (3.6%), as shown in Figure 1.” (lines 190-192)

  1. Explain figure 3, the meaning of axis 1, 2 and 3.

It was modified accordingly. “Axes 1, 2, and 3 explain 9.461%, 5.425%, and 4.839% of the variation, respectively. The distribution of samples shows a partial separation by surface type, suggesting that surface characteristics may influence bacterial composition after cleaning”.

  1. It would be good to further specify how many samples were not tested due to lack of DNA, and how many due to failure to meet the bioanalysis criteria? Which bioanalysis criteria were not met?

The manuscript was rephrased: “The excluded samples from the poultry slaughterhouse included 20 (40%) samples with no colony growth on PCA plates and 7 (14%) samples, of which 6 had insufficient DNA for sequencing and 1 failed due to poor read quality.” (lines 175 and 176)

  1. It would be useful to list which samples - from which surfaces – were bacteriologically negative after incubation and to try to explain.

It is now indicated in Table 2.

Table 2. Presence and absence of bacterial growth on PCA plates from samples evaluated on equipment surfaces in Brazilian poultry and fish slaughterhouses.

Collection points

N +

%

N -

%

Poultry slaughterhouse (n=50)

Modular polypropylene conveyor belt

4

8

6

12

Modular polypropylene conveyor belt

7

14

3

6

Stainless steel table

6

12

4

8

Smooth polyurethane conveyor belt

7

14

3

6

Modular polypropylene conveyor belt

6

12

4

8

Fish slaughterhouse (n=50)

Modular polypropylene conveyor belt

6

12

4

8

Smooth polypropylene cutting board

5

10

5

10

Skin machine (stainless steel)

4

8

6

12

Modular polypropylene conveyor belt

5

10

5

10

Modular polypropylene conveyor belt

8

16

2

4

(+) indicates the number and percentage of samples with bacterial growth on PCA plates.

(-) indicates the number and percentage of samples with no bacterial growth on PCA plates.

Authors: thank you.